# Impact of natural disasters on consumer behavior: Case of the 2017 El Niño phenomenon in Peru

**Hugo Alatrista-Salas**[1], **Vincent Gauthier**[2]*, **Miguel Nunez-del-Prado**[1]*, **Monique Becker**[2]

**1** Universidad del Pacífico, Lima, Peru, **2** Laboratory SAMOVAR, Telecom SudParis, Institut Polytechnique de Paris, Palaiseau, France

* vincent.gauthier@telecom-sudparis.eu (VG); m.nunezdelpradoc@up.edu.pe (MN)

**Data Availability Statement:** The dataset used was obtained from a Peruvian private financial entity. The dataset was provided to us in the context of a long-standing collaboration between the

## Abstract

El Niño is an extreme weather event featuring unusual warming of surface waters in the eastern equatorial Pacific Ocean. This phenomenon is characterized by heavy rains and floods that negatively affect the economic activities of the impacted areas. Understanding how this phenomenon influences consumption behavior at different granularity levels is essential for recommending strategies to normalize the situation. With this aim, we performed a multi-scale analysis of data associated with bank transactions involving credit and debit cards. Our findings can be summarized into two main results: Coarse-grained analysis reveals the presence of the El Niño phenomenon and the recovery time in a given territory, while fine-grained analysis demonstrates a change in individuals' purchasing patterns and in merchant relevance as a consequence of the climatic event. The results also indicate that society successfully withstood the natural disaster owing to the economic structure built over time. In this study, we present a new method that may be useful for better characterizing future extreme events.

## Introduction

El Niño–Southern Oscillation (ENSO) is a climatic phenomenon consisting of a temperature increase in the equatorial Pacific area. ENSO has a 2-7 years fluctuation period, with a warm phase known as El Niño and a cold phase known as La Niña. A crucial indicator of the presence of El Niño is the variation of the sea surface temperature, which causes changes in the worldwide climate. At the end of 2016 and in early 2017, ENSO had an abrupt change that caused heavy rains and floods. This atypical phenomenon is called El Niño costero. According to United Nations Office for the Coordination of Humanitarian Affairs (OCHA) [1], the first three months of 2017 witnessed the highest amount of human and material loss in Lima and in the northern regions of Peru caused by the coastal ENSO phenomenon. In this paper, we focus on two main events that occurred in February and March 2017 (see Fig in [2]).

In February 2017, strong rainfall accumulation led to 39 fatalities, 14 injured individuals, 8,299 affected individuals, 19 destroyed bridges, 29 affected bridges, 11.92 km of damaged

Universidad del Pacífico and the financial entity through a specific multidisciplinary research agreement. Sharing the raw version of this dataset can potentially breach the privacy of the bank's customers. However, the minimal underlying dataset necessary to replicate the study's findings is publicly available from https://doi.org/10.7910/DVN/LYXBGR.

**Funding:** VG, MB: The 3rd Programme d'Investissements d'Avenir ANR-18-EUR-0006, NO, https://anr.fr/ VG, HA, MNP: The STIC AM-SUD program through the 04-2017 PEDESTAL project, NO, http://www.sticmathamsud.org/

**Competing interests:** The authors have declared that no competing interests exist.

roads, 140.39 km of affected roads, 191.5 ha of destroyed crops, 1,472 ha of affected crops. In addition, the northwest region of Peru and the southern Arequipa region were both in a state of emergency. The second event in March 2017 inflicted more damages on the country, leading to 98 deaths, over 1 million affected individuals, 639 affected bridges, 1,722 affected schools, 351 injured individuals, 20 missing individuals, 605 affected hospitals, 8,481 km of affected roads, 230,317 damaged houses, and 5,244 ha of deteriorated crops, as illustrated in Fig in [3].

In this work, the goal is not to estimate the macroeconomic impact of extreme climatic events such as in [4, 5], but to better understand how resilience leads a population to organize itself after a shock through the prism of the population purchasing behavior. In recent years, in Peru, the El Niño phenomenon has harmed the economy of Peru due to the damage it caused to the country's infrastructures. However, it has also directly impacted the economic life of Peruvian citizens. For instance, the last event of El Niño costero in 2017, led to rising food prices of lemon and garlic, which are two basic foodstuff in the Peruvian diet. The availability of food supplies was generally due to the degradation of the road network. The degradation of the infrastructure also made it difficult to supply, water, vegetables, and meat to major cities. The unavailability of food items triggered panic buying of the missing items throughout retail stores in impacted and non-impacted areas. It remains unclear how these reported events impacted the consumption behavior of people. In addition, the dynamics of individual purchases during a time of crisis remain poorly studied, which gives us further motivations to study purchase behavior from a time perspective during the transient period of the El Niño event of 2017.

In this study, we aimed to determine the resilience of retail structures by measuring the collective response of consumers living in Lima's greater area. In particular, we developed our analysis to better understand the consumer habit changes during a period of climatic stress. To achieve this goal, we performed a multi-scale analysis of the consumption patterns based on a credit and debit card transaction dataset of roughly 6 million Peruvian citizens gathered over a 2-year period from 2016 to 2017. In this study, we focused exclusively on the city of Lima for two reasons. First, it is both the political and economic capital of Peru, containing roughly 1/3 of the country's population. Second, despite the fact that the data are available for the entire country, data from the other regions are sparse due to the lack of bank coverage in certain areas of the country, which makes it difficult to have even coverage of the country.

We first focus on techniques to measure the dynamics of consumer behavior at the macroscopic level to evaluate our central hypothesis that consumer behavior shifted in the aftermath of the 2017 ENSO events. This phenomenon impacted the city of Lima twice: once in mid-February and once at the end of March. We demonstrate that other shocks did not impact consumer behavior as much as these two events.

At the regional level, we examined people's consumption patterns through individual mobility models. We observed that purchase behavior patterns changed as a consequence of the ENSO, but in a non-homogeneous way. We also examined how specific merchants responded during the events. Our main finding was that despite the fact that the overall economic activity slowed in response to the events, businesses responded differently during the events and in their aftermath. In this paper, we aim to improve preventative measures that can mitigate climatic events and improve the effectiveness of recovery efforts. Our contributions are summarized as follows:

1. We captured anomalous events using the Kullback-Leibler divergence (KLD). The main purpose was to recognize a significant change in the purchase distribution of the population over time as an indicator of an anomalous event.

2. We measured the changes in people's behavior using the mobility Markov chain (MMC) model. The basic principle was to quantify changes in both individual purchase categories and frequent locations.

3. We quantified how individual merchants reacted during an event by studying the evolution of the PageRank of each merchant in the transaction graph. Here, using the PageRank enables us to characterize how the attractiveness of a merchant compares to others.

4. We measured the evolution of the core/periphery structure of the transaction graph during the events.

In facing extreme climatic events, resilience has emerged as a key concept for understanding how communities and systems are able to absorb and adapt to stress and shocks [6, 7]. Recently, the National Academies of Sciences, Engineering, and Medicine released a study [8] on strengthening supply chain resilience in the aftermath of hurricanes. Four keys domains were identified that must be maintained to foster the resilience of society: power and communications networks, food and water supply, fuel supply, and medical and pharmaceutic supply. Resilience is generally studied with respect to the ecosystem, and few works [9–13] have explored and analyzed the resilience of social systems facing extreme climatic events. Wang *et al.* [9] and Guan *et al.* [11] studied Hurricane Sandy through the lens of human mobility perturbation. Bagrow *et al.* [14] explored the societal response to external disturbances, such as bomb attacks and earthquakes, by studying mobile phone communication patterns. Niles *et al.* [12] and Eyre *et al.* [15] studied social media usage during a climatic event. In particular, they found differences in tweet volume for keywords depending on the disaster type, with people using Twitter more frequently in preparation for hurricanes and for real-time recovery information on tornado and flooding events. With the same goal, in [13], the authors analyzed emotion-exchange patterns that arise from Twitter messages sent during emergency events. Additionally, in a joint work by Banco Bilbao Vizcaya Argentaria (BBVA Data & Analytics: https://www.bbvadata.com/) and UN Global Pulse (UN Global Pulse: https://www.unglobalpulse.org/), Martinez *et al.* [10] analyzed bank debit and credit card payments and ATM cash withdrawals to map and quantify how individuals were impacted by and recovered from hurricane Odile.

The following works [16–19] analyzed customer behavior by studying the sequence of purchases through credit and debit cards. For instance, in [16], the authors used the Sequitur algorithm [20] to classify user spending behavior and characterize people's lifestyles according to their temporal purchase sequences. In addition, Leo *et al.* [17, 18] used a detection algorithm to characterize people purchase sequences characterized by the merchant category code (MCC) [21]. Finally, another model based on retail customer data [19] identified temporal regularities in buying behavior. The authors grouped weekly customer buying patterns using a k-means clustering algorithm to extract groups of behaviors $G_u$ per user.

Instead of using debit and credit card data to characterize user spending behavior, in [22, 23], the authors attempted to characterize cities based on the economic activity of their residents. Youn *et al.* [22] created a model to predict how individual business types systematically change as the city size increases, shedding light on the processes of innovation and economic differentiation. To build the model, the authors used the National Establishment Time Series dataset. The authors used approximately 50 million shopping transactions of 91,000 customers between January 1, 2007 and June 1, 2015 in Leghorn province, Italy. In [23], the authors demonstrated that urban socioeconomic quantities and individual spending activity scaled superlinearly with city size. The approach was assessed through bank card transactions of both debit and credit cards of Spanish clients of Banco Bilbao Vizcaya Argentaria (BBVA) with a dataset containing 178 million transactions made by 4.5 million clients in 2011.

To the best of our knowledge, no research thus far has been conducted to capture the fine-grained impact of extreme climatic events on the spending behavior of individuals and small businesses. In this study, we aim to provide a deeper understanding of retail distribution in the aftermath of a natural disaster.

## Results

### Kullback-Leibler Divergence (KLD) analysis of the bank transaction distribution

To investigate how the ENSO events in February and March 2017 impacted consumption patterns in Lima, we examined how the relative frequency of merchant categories evolved in time using the KLD. In addition, we demonstrate in Fig 1(a) that customers' indeed slowed significantly both in number and volume during the first event of February 2017. Furthermore, a smaller number of transactions was observed during the second event in March, but the magnitude was less than that of the February event. The same behavior was observed for cash withdrawal (see Fig 1(b)).

As a reference, Fig 1c illustrates the distribution of the share of spending in each category of purchases (VISA Merchant Category Classification MCC) in our dataset. The figure displays only the 50 most consumed categories throughout the country averaged throughout our dataset. It can be observed that the distribution of the frequency of each category of purchases follows a Zipf-like distribution with dominant purchases in for food-related stores (*i.e.*, grocery stores and supermarkets), as expected.

To further explore the evolution of the consumption pattern in response to events, we used two different divergence measures $\mathcal{D}^{(1)}$ and $\mathcal{D}^{(2)}$ to quantify the deviation of the purchase behavior at a given time from the normal purchase behavior. To compute the purchase distribution we classified each purchase record with its MCC information into 15 different categories based on the Classification of Individual Consumption According to Purpose (CIOCOP), as displayed in Table 2. With the divergence measure $\mathcal{D}^{(1)}$ (see Fig 2(a)) defined in (2), we measured the divergence of the distribution of the purchase behavior $\mathbf{S}_i^{(t)}$ (made in district $i$ of the greater Lima area during the interval $[t, t + w]$) from the average consumption behavior of the entire country in each purchase category $\bar{\mathbf{S}}$. With the divergence $\mathcal{D}^{(2)}$ (see Fig 2(b)) defined in (3) we used a slightly different approach by computing the average KLD between $\mathbf{S}_i^{(t)}$, the purchase behavior in district $i$ at time $t$ and the purchase behavior at a reference date.

The metric $\mathcal{D}^{(1)}$ displays a smoothed evolution of the KLD highlighting the macroevolution, while metric $\mathcal{D}^{(2)}$ displays a more detailed the evolution of the KLD, including characteristic weekly behavior with relatively stable behavior weekdays and different behavior during weekends.

In Fig 2(b) and 2(C), we present the average divergence of all districts of Lima. We can observe that with both divergence measures $\mathcal{D}^{(1)}$ and $\mathcal{D}^{(2)}$ the ENSO events in mid-February and mid-March appear very distinctly. This suggests that purchases made in certain categories temporarily shifted toward other categories in response to the ENSO events. In Fig 2(a) we show the 7-day cumulative rainfall over Lima city and the water catchment area of Lima rivers. The February event was the result of a long period of continuous rainfall both over Lima city and the water catchment area of Lima's rivers that has tipped around the mid-February and has induced flooding and damages. The March event was characterized by a short period of intense rainfall over Lima and its water catchment. In S2 Fig in the S1 File, we present the evolution of the KLD at the district level and observe that not all districts responded evenly to the events.

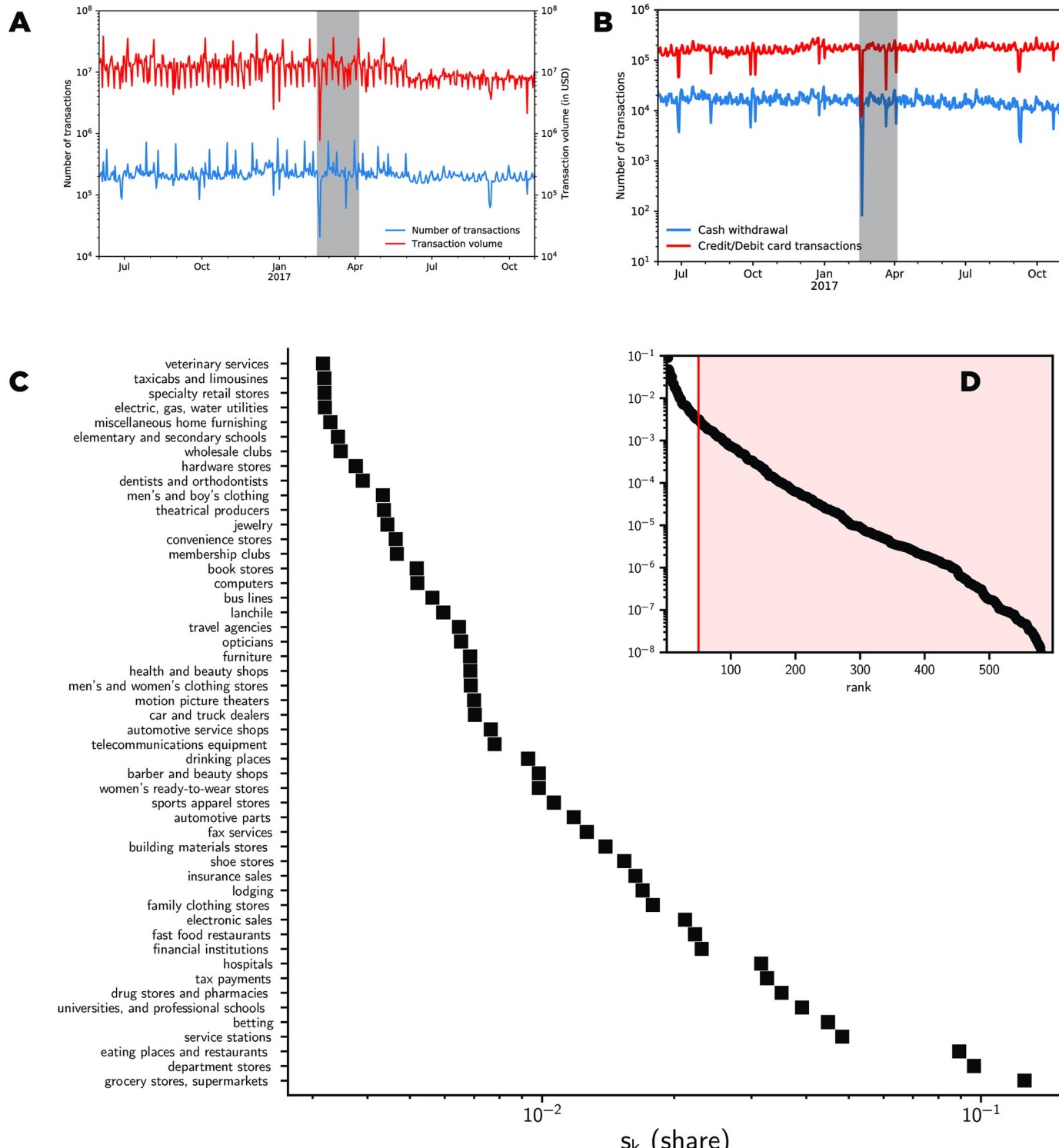

**Fig 1. Bank transaction time series and distribution. a**) Transaction volume (red) and frequency (blue) in time. **b**) Transaction frequency $S_k$ by type, where the transaction type is defined by the merchant category code (MCC) of a merchant (only the 50 most frequent MCC codes are displayed). **c**) Full distribution of the MCC distribution.

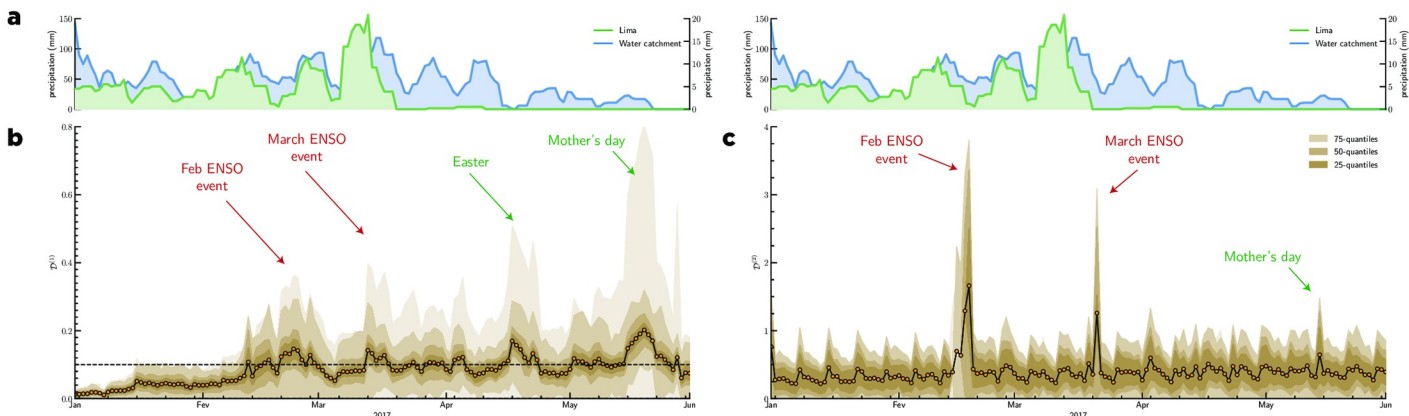

**Fig 2. Divergence $\mathcal{D}^{(1)}$ and $\mathcal{D}^{(2)}$ of various districts of Lima.** The average divergence over all districts of Lima as well as the 25th, 50th, and 75th quantiles are plotted in different shades of brown. **a)** sliding window of 7 cumulative rainfalls over Lima city (in green) and the water catchment area of Lima's rivers (in blue); the right y-axis represents 7 days cumulative rainfall over the water catchment area of Lima's rivers (blue) and the left y-axis represents the 7 days cumulative rainfall over the area of Lima (green); the data were obtained from the Center for Hydrometeorology and Remote Sensing (CHRS) Data Portal [24]. **b)** Divergence $\mathcal{D}^{(1)}$. **c)** Divergence $\mathcal{D}^{(2)}$.

### Causality analysis of the ENSO over the individual purchasing behaviors

To determine whether all district of Lima were affected by the event uniformly, we performed a causal impact analysis [25] on the purchase patterns in each district of Lima after the first event in February. We discovered three modes: districts that were negatively impacted, districts that continued to function as usual, and districts that experienced an increase in purchases. Fig 3 presents the causal impact analysis for 42 districts of Lima, using the Callao series as s control. As a result, three different effects of El Niño can be observed. Fig 3(a) is an example of a decreasing trend in the post-intervention time, displaying a negative impact after the appearance of El Niño. These districts include *Lima, Cieneguilla, San Martin de Porres, Ate, San Juan de Lurigancho, Pucusana, Lurigancho, Los Olivos, Ancon, Chorrillos, Santa Rosa, San Bartolo, Jesus Maria, Surquillo, Santa Maria del Mar, Villa el Salvador, Punta Hermosa, Lince, Lurin*, and La Victoria. In contrast, the increasing trend in Fig 3(b) reveals a positive impact of El Niño in *Pachacamac, Carabayllo, San Isidro, San Borja, Santa Anita, El Agustino, Rimac, Santiago de Surco, Pueblo Libre, Breña*, and *Punta Negra* districts. In addition, Fig 3(c) displays a neutral effect in the post-intervention, which signifies that El Niño did not significantly affect the remaining districts. To summarize the results Fig 3(d) illustrates the results of the decreasing, increasing, and stable trends in green, yellow and red, respectively. We note that the affected districts are close to the Huaycoloro, Chillón, Lurín, and Rímac rivers causing floods.

To better understand the causal effects, we query to an official report issued by Instituto Nacional de Defensa Civil (INDECI), the Peruvian organism in charge of Disaster Risk Management, on February and March 2017, heavy rainfalls were recorded in several Lima districts, including San Juan de Lurigancho, Lurigancho-Chosica, Chaclacayo, Pachacamac, and Comas. These rains caused numerous floods due to the overflow of rivers Huaicoloro, Lurín, Rimac, and Chillón. The INDECI report also mentions that many businesses were closed during the period, and the inhabitants of these areas were forced to make their purchases in other districts to buy food and other needs. So, these districts show a negative impact. We also hypothesize that the districts with a positive impact have received an influx of purchases from the most impacted districts.

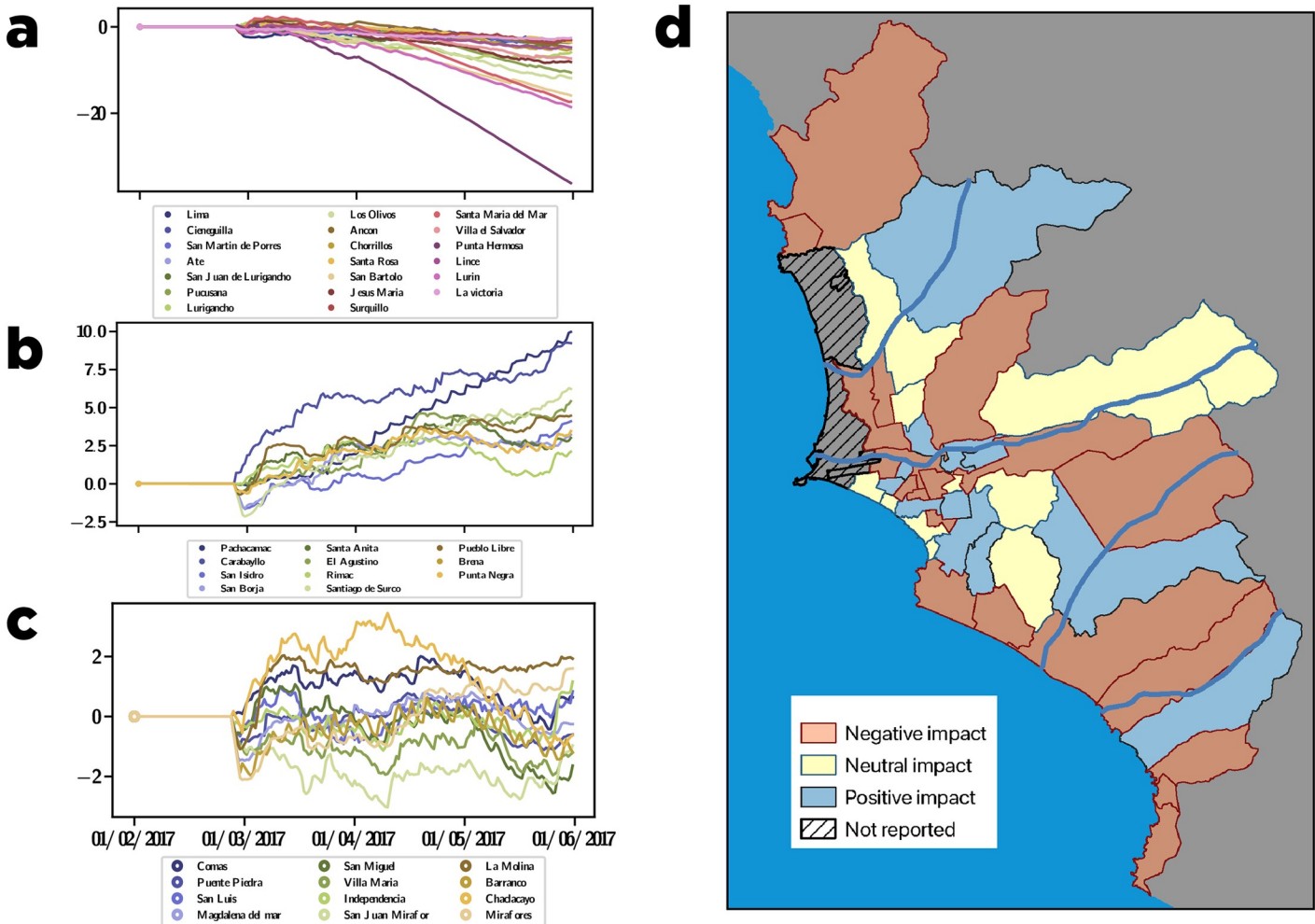

**Fig 3. Causal impact at the districts level for February 2017. a**) List of districts that were negatively impacted. **b**) List of districts that were positively impacted. **c**) List of districts that experienced a neutral impact. **d**) Map of Lima showing the districts with a negative (red), positive (green) and neutral (yellow) impact.

## Individual purchasing behavior

In this subsection, we focus on the impact of the El Niño phenomenon on people through individual MMC as a proxy for whether an individual was affected. The official Peruvian definition refers to a person, animal, territory, or infrastructure suffering disturbance in its environment due to the effects of a phenomenon. Immediate support may be required to reduce the effects of the disorder by continuing regular activity [26]. Under normal conditions, people tend to buy items from the same categories, such as "Food and non-alcoholic beverages", "Clothing and footwear", and "Transportation". However, in the presence of a disruptive phenomenon, frequent purchase categories can change. Thus, the stationary vector of the MMC model represents the probability of buying from a given merchant and therefore from the category. It should be noted that merchants belonging to the same purchase category are merged, and their respective probabilities are added. Finally, the categories are sorted according to their weights.

With respect to the variation in purchase categories for an individual *i* over time, we used four weeks of individual historical consumption data to compute the stationary vector $\pi_t$.

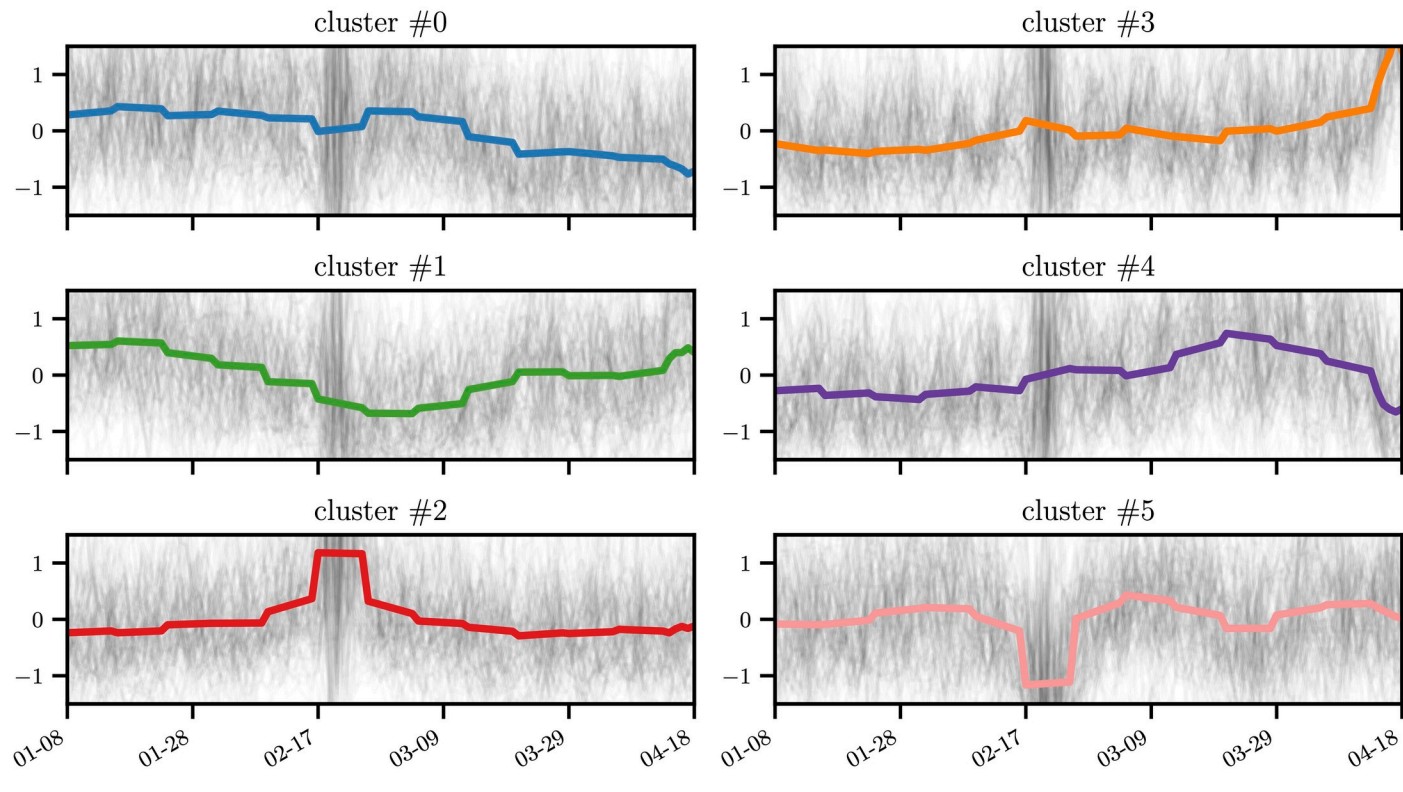

**Fig 4. Time series clustering.**

Thus, we built a set of consumption stationary vectors $csv_i = \{\pi_t, \pi_{t+1}, \ldots, \pi_{t+n}\}$ shifted by one week for individual $i$. Accordingly, we used the normalized discounted cumulative gain metric (see Fig 4) to measure the variability between two consecutive consumption patterns $\pi_t$ and $\pi_{t+1}$ belonging to individual $i$. Finally, we averaged the variations for all individuals living in different districts of Lima.

Fig 5 reveals a change in purchase patterns in all districts on approximately February $15^{th}$. First, the buying categories changed noticeably in residential areas compared to vacation property districts, such as Cieneguilla, Santa Rosa, San Bartolo, and Pucusana. Second, in residential districts, people could either not reach gas stations, or gas stations suffered a fuel shortage due to infrastructure degradation. Therefore, individuals tended to use more public transportation services, or ridesourcing services such as Uber, and Cabify. There was also an increase in purchases in the insurance, home furniture, and health categories. Finally, vacation property districts demonstrated an increase in the health and technology categories, while the clothing category decreased.

## Merchant network resilience

It is well known that the distribution of the purchases is a highly skewed distribution (Zipf-like distribution) toward certain purchase categories [16, 27], where food-related categories are the the dominant categories (see Fig 1). With such a highly skewed distribution, it may not always be easy to detect variations in the empirical distribution due to the fact that certain categories are hidden in the tail of the distribution. Even if the divergence metric displays a clear sign of a shift in the distribution, to avoid these pitfalls, we used a different approach and analyzed the

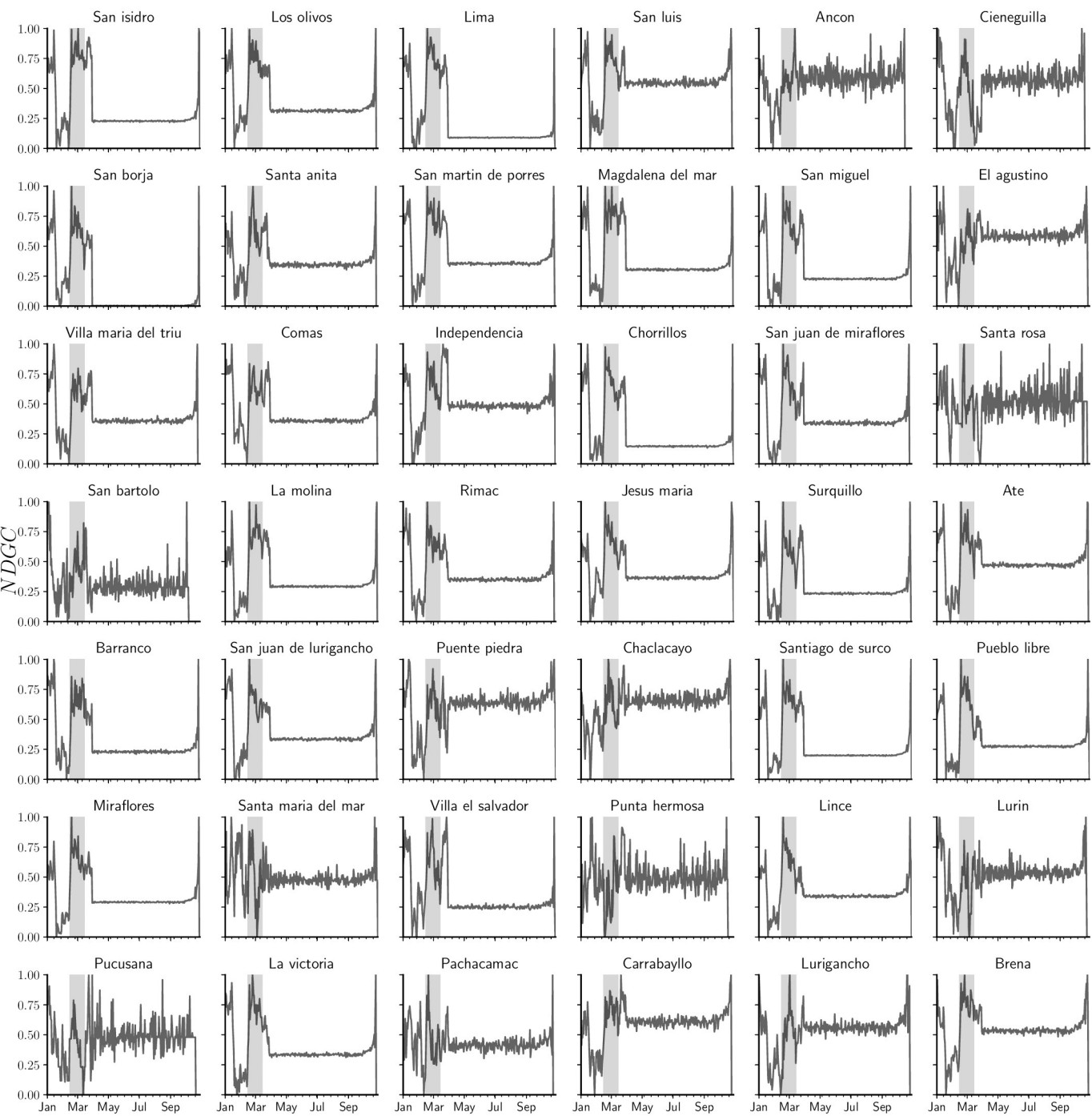

**Fig 5. Average variation of purchase category composition.** Stationary vectors of the mobility Markov chain models were used as input and were built from four weeks of consumption by a week for the normalized discounted cumulative gain (NDCG) gain for all individuals living in a given district.

microscopic dynamic of each merchant during the ENSO events. Specifically, we analyzed the merchant dynamics during the ENSO events, studying the ranking evolution of individual merchants through the analysis of the discrete evolution of their PageRank in the transaction graph. Additional information on the preprocessing of the transaction graph is provided in the

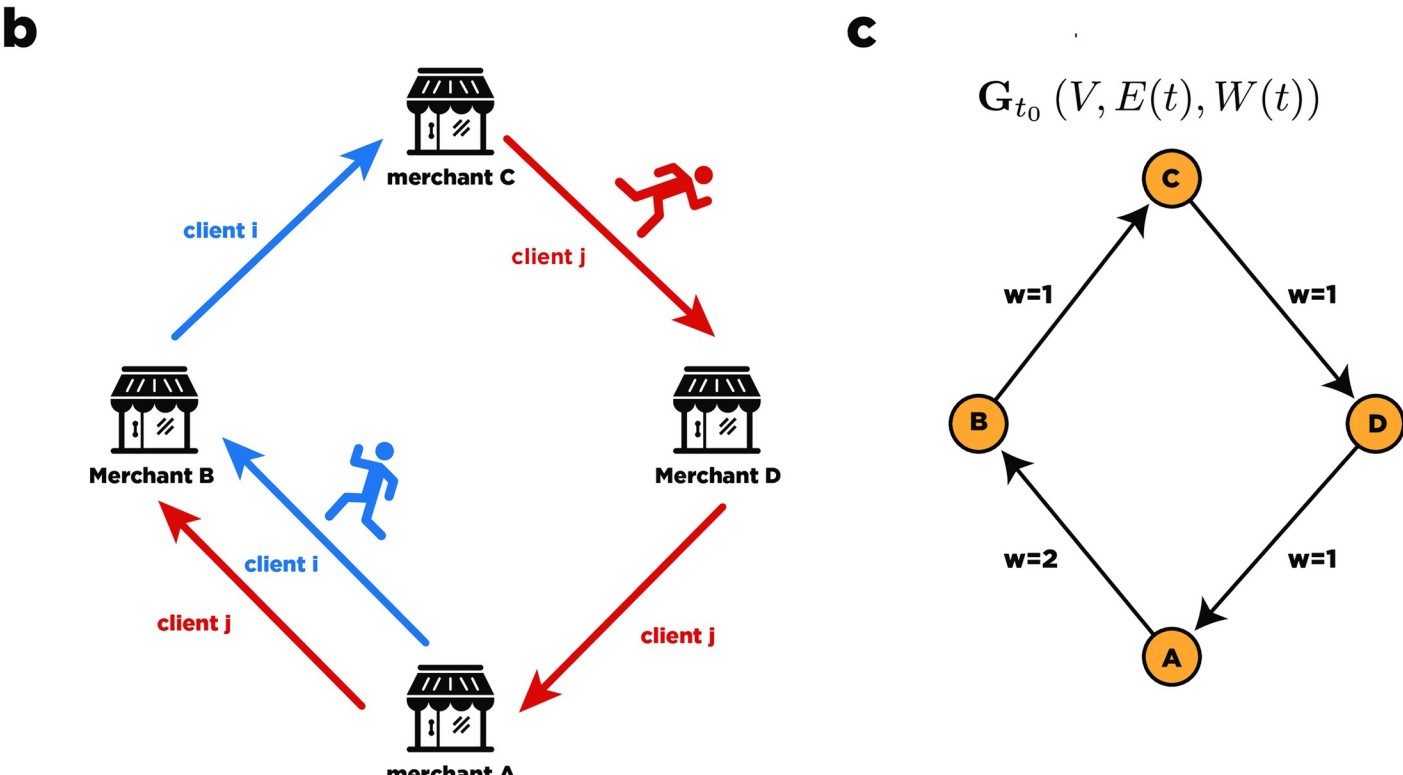

**Fig 6. Transactions graph. a**) Purchase sequences **b**) Users purchase sequence represented as directed graph. **c**) Users purchase sequence represented as directed Weight graph, where the weights represent the number of transactions made during a given time slice between two merchants.

PageRank section. The transaction graph (see Fig 6) is an aggregation of the transaction records into a weighted and directed graph. Based on the transaction graph, we computed the PageRank of the nodes in the resulting directed and weighted graph.

To analyze the ranking evolution of each merchant, we split the transaction graph into time slices $\mathcal{G} = [\mathbf{G}_{t_0}, \mathbf{G}_{t_1}, \mathbf{G}_{t_2}, \ldots]$. Computation of the PageRank for each time slice enabled us to create a time series $\mathbf{r}_i(t)$ that represents the temporal evolution of the ranking of each merchant $i$. In Fig 7, we provide several examples of the PageRank evolution for different merchant categories. Finally, we clustered the time series $\mathbf{r}_i(t)$ using the kmeans algorithm once we transform each the time series using a symbolization technique for time series (1d-SAX). Additional information on the clustering method is provided in the Method section. As a result, six

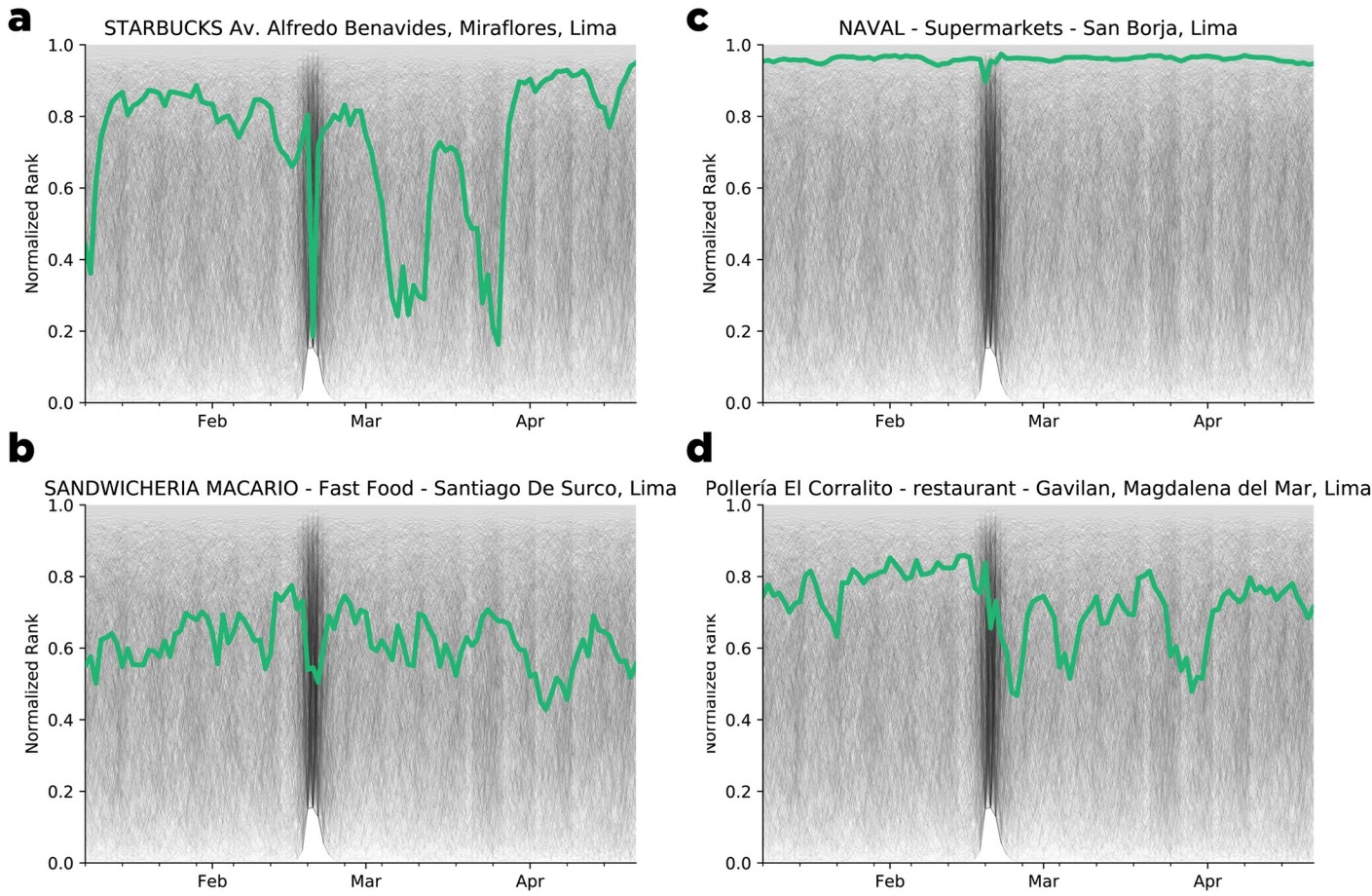

**Fig 7. Example of the normalized rank evolution of merchants in Lima, Peru, during the interval of January 2017 to April 2017.** a) Starbucks coffee shop. b) Fast food restaurant. c) Supermarket. d) Restaurant.

distinct profile patterns (see Fig 4) emerged that highlighted different response profiles as a function of the merchant category and area during the ENSO events.

In Fig 8(a), we illustrate the distribution of the merchant categories in each cluster. Our main findings are summarized in Fig 9. First, The cluster #5 is a cluster of merchants that experienced a drop in their ranking during the ENSO events in February. Here, we observed an overrepresentation of gas stations and food-related merchants. In contrast, for cluster #2, where merchants experienced a surge in their ranking during the ENSO events in February, we observed an overrepresentation of health related merchants and gas stations. These results can be explained as follows: some gas stations experienced shortages, and there was a surge in the demand for health-related products.

In Fig 8, two additional phenomena can be observed: first, an increase in insurance purchases after the first event in February (cluster 3), and second, an increase in purchases of technology related items during the first event (cluster 2). We observe a surge in the purchased of new insurance policies after the first event. It should be noted that Peru is an underinsured country (only approximately three in hundred houses are insured), and in the aftermath of the event it appears that people decided to purchase new insurance policies. With respect to technology-related items, a query to the database at our disposal revealed that no transaction were

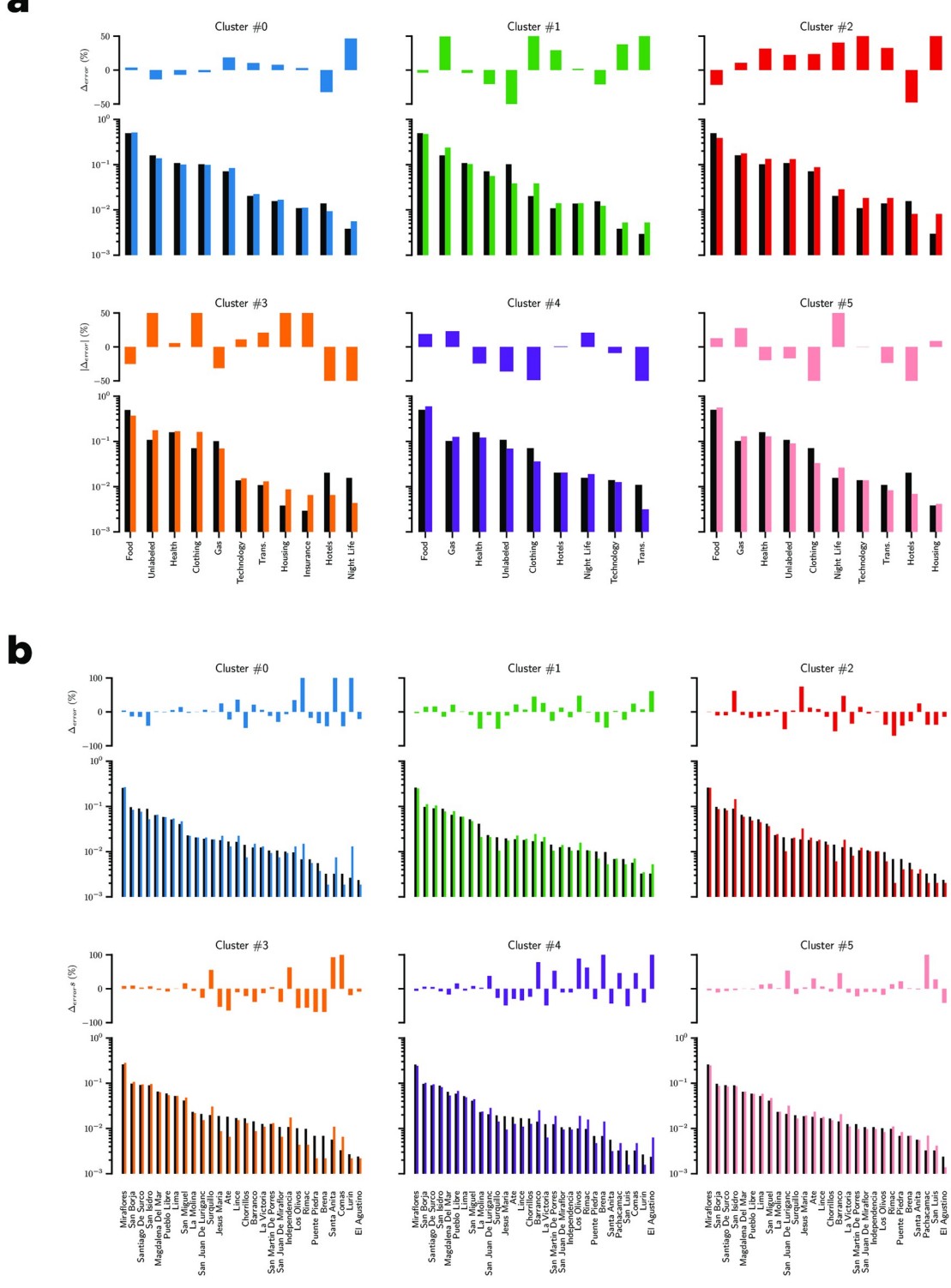

**Fig 8. Proportion of merchant categories/area inside different clusters. a)** Proportion of merchants per cluster as a function of their Classification of Individual Consumption According to Purpose (COICOP), **b)** Proportion of merchants per cluster as a function of their district area. The top of each figure indicates the relative differences between the proportion of a given category inside a cluster and the proportion of that category in our dataset.

| cluster | ranking | over represented categories | under represented categories |
|---------|---------|------------------------------|------------------------------|
| #0 | | health | food |
| #1 | | gas, clothing | unlabeled stores |
| #2 | | health, gas, technology, transports | food |
| #3 | | health | food |
| #4 | | food, gas | health, clothing |
| #5 | | gas, food | clothing, night-life |

**Fig 9. Summary of most important merchant categories found in each cluster.**

made in that category from February 15 to 19, 2017. We observed a shift in purchases after February 19. This behavior appears to be due to the purchase of prepaid cell phone plans.

We note that people's responses to the crisis were very heterogeneous in time, space and behavior. We believe that the microscopic approach developed in this study is a key methodological innovation that was helpful for fully understanding the extent of a crisis and the societal outcome of an event.

## Resilience transaction graph

The presence of a core/periphery structure in networks is an indication that the overall network structure is resilient to the random removal of some nodes [28, 29]. By exploring the transaction graph, we observed that a very small core of merchants (constituted of less than 1% of merchants) was almost fully connected and surrounded by a vast periphery that was connected to the core in a tree-like manner. The merchants belonging to the core structure were often large supermarkets that provide a vast array of goods including basic necessities. To monitor the size of the core network structure over time, we computed the temporal evolution of the size of the core structure of the transaction graph at each time slice $\mathcal{G} = [\mathbf{G}_{t_0}, \mathbf{G}_{t_1}, \mathbf{G}_{t_2}, \ldots]$. To derive the size of the core structure, we used the method developed by Ma *et al.* [30] (see Materials and methods section for more detailed information), a method designed to detect the core/periphery structure in a directed and weighted graph. In the S1 File, we also provide the results that exploit a different approach with the k-core decomposition algorithm. Our main findings are summarized in S4a Fig in S1 File we explore the dynamics of the transaction graph. In S4b Fig in S1 File we see that the node that belongs to higher core concentrates a higher proportion of transactions than nodes in lower cores.

As previously demonstrated, ENSO events impacted the buying people's buying patterns; however, in Fig 10 we also demonstrate that these events significantly impacted the size of the core structure $|V_c(t)|$ of the transaction graph. Our main finding is that the core size distribution is similar to the normal distribution. We calculated the Kolmogorov-Smirnov (KS) distance between the core size distribution $|V_c(t)|$ and normal distribution $\sim \mathcal{N}(68, 17)$, and the KS-test produced $\mathcal{D} = 0.09$ and p-value = 0.19. Nonetheless, despite the relative accordance with the normal distribution, the core size distribution shows in Fig 10b demonstrates extreme values at both tails that deviate from the normal distribution. In Table 1, we display the list of noticeable events with their standard scores. Our results reveal that for both ENSO events in mid-Feb and mid-March, the core size significantly decreased from the usual behavior compared to Mothers' Day, or Easter (two particularly popular holidays in the Peruvian culture)

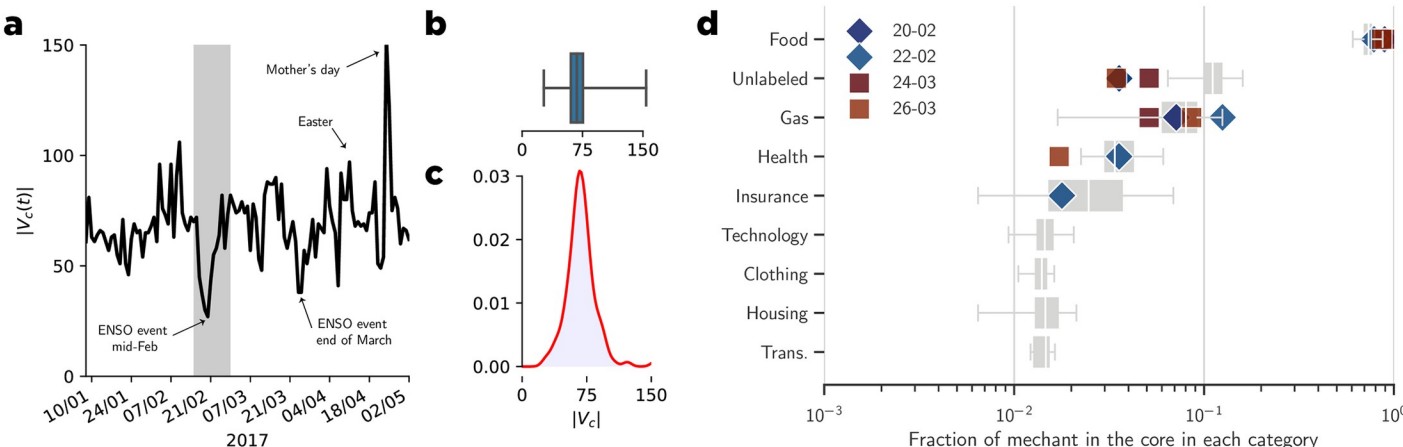

**Fig 10. Evolution of the core structure of the transaction graph over time. a)** Temporal evolution of the size of the core structure of the transaction graph. **b)** Boxplot of the core size distribution. **c)** Core size distribution approximated by kernel density estimation. **d)** Fraction of merchants that belong to the core nodes split by categories, the diamond represents the category split during the first ENSO event of February, whenever the square represents the second event of end of March.

where the core size significantly increased. In Fig 10(d) we see the changes in the category distribution of merchants present in the core during both events (the February event and the March event). During these events only a subset of the merchant's categories remain present in the core. Health and food related merchants are the main remaining categories.

## Discussion and conclusion

In this study, we explored the impact of the ENSO events of 2017 on retail sales through the lens of a massive transaction dataset from the greater area of Lima, Peru to understand how the population handled the aftermath of the climatic events. At the macroscopic level, despite a clear slowdown of economic activity triggered by the two main ENSO events occurring in February and March, we demonstrated that the overall economic activity recovered swiftly from the events (see Fig 1). The second event appears to have had less impact on the economic activities than the first event, although its intensity was not smaller. A more detailed analysis of the events indicated that regions that registered more damages induced by the events also suffered from a long-term deficit of consumption compared with other areas (see Fig 3). We quantified how individual purchase categories and frequent locations changed during the events (see Fig 5), demonstrating that there was a transient period during which people changed their purchase sequences to accommodate new necessities or constraints. By tracking the ranking evolution of each small business in the transaction graph, we revealed that small businesses were impacted very differently based on their category. A subset of merchants, such

**Table 1. Standard score of core size $|V_c(t)|$ at various event times.**

| date | $|V_c(t)|$ | z-score | event type |
|---:|---:|---:|---|
| 2017/02/20 | 27 | -2.44 | ENSO event mid-Feb |
| 2017/03/24 | 38 | -1.79 | ENSO event end of March |
| 2017/04/07 | 41 | -1.61 | ENSO event beginning of April |
| 2017/04/11 | 97 | 1.67 | Easter vacation |
| 2017/05/17 | 154 | 5.02 | Mother's day |

as pharmacies, hospitals, gas stations, and grocery stores exhibited a surge of activity during the events. In contrast, merchants that sold non-necessities experienced a drop in their ranking during the events. In addition, by studying the core network structure of the transaction graph, we observed a clear reduction in the size of the core network structure during both events, which contrasts with other types of events, such as Easter or Mother's Day, where the core structure increased in size.

We analyzed different small subsets of merchants in various districts to evaluate alternative explanations for the slowdown of sales, such as failure of point-of-sale systems and problems in payment architectures. However, none of the analyzed merchants experienced these problems. Nevertheless, almost all merchants encountered severe problems with the water supply during the events. We also asked merchants about the decrease in the number of clients during the two events of the El Niño phenomenon in 2017. The responses were the same: the number of customers did not decrease, and the usual number of employees attended. We believe that this was due to the fact that the vital infrastructure in inner Lima did not suffer significantly during the event.

Despite the strength of the event, Peruvian society continued its activities due to the economic structure and the manner in which the population faced the event. One explanation is that Peruvian society is used to periodic climatic events that occur on a 2–4 year basis, although the 2017 event was stronger in intensity.

We believe that beyond the case of the ENSO phenomenon, the methodological tools designed and developed in this study can help further understand the microscopic dynamics that underpin the societal outcome of climatic events. We believe that our approach based on the microscopic analysis of consumption patterns can help build information systems to aid the population during the relief effort period of a climatic event or other type of societal shock. However, this microscopic approach may raise privacy leaks [31] that need to be further addressed by applying privacy techniques such as in [32, 33], while minimizing the impact of privacy techniques on the relevance of our study. Moreover, lots remain to be learned on how we react as a human and as a society during a crisis, such as an extreme climatic event, this field remains open for further inquiry. We think, for instance, that it would be interesting to separately estimate the impacts for cohorts with different social-economic factors; or if a person is impacted differently based on his/her gender or age.

## Materials and methods

### Transaction dataset

This dataset was gathered from June 2016 to May 2017, containing approximately 1.5 million clients, 55,000 distinct merchants, and 116.8 million transactions from both credit and debit cards in Peru. These data are associated with customer consumption registered by credit and debit cards in stores located in Peru.

The dataset is composed of the following features:

1. Features describing the clients such as anonymous ID, age, gender, and country in which the card was issued.

2. Features describing the transaction, such as the timestamp, amount spent in Peruvian currency, and the number of transactions.

3. Features associated with the bank agency, namely the region, province, and district, in which the agency of the client was located.

**Table 2. Classification of Individual Consumption According to Purpose (COICOP).**

| Code | Name |
| --- | --- |
| 01 | Food and non-alcoholic beverages |
| 02 | Alcoholic beverages, tobacco and narcotics |
| 03 | Clothing and footwear |
| 04 | Housing, water, electricity, gas and other fuels |
| 05 | Furnishings, household equipment and routine household maintenance |
| 06 | Health |
| 07 | Transport |
| 08 | Information and communication |
| 09 | Recreation, sport and culture |
| 10 | Education services |
| 11 | Restaurants and accommodation services |
| 12 | Insurance and financial services |
| 13 | Personal care, social protection and miscellaneous goods and services |
| 14 | Individual consumption expenditure of non-profit institutions serving households |
| 15 | Individual consumption expenditure of general government |

4. Features characterizing the merchants, such as merchant ID, merchant name, merchant address, the MCC [21] and the Lambert coordinates of the merchants.

In this study, we merged the MCC categories into a more meaningful categorization, often used in microeconomics [34], namely, Classification of Individual Consumption According to Purpose COICOP [35]. The COICOP aims to divide individual consumption expenditures into 15 categories, as depicted in Table 2.

## Kullback–Leibler Divergence (KLD)

To analyze buying patterns, we used the KLD defined in (1) to compute the two different divergence measures $\mathcal{D}^{(1)}$ and $\mathcal{D}^{(2)}$. The KLD is a general measure of dissimilarity between two probability distributions. In (2), we define $\mathcal{D}^{(1)}$ as the KLD between the probability distribution of the share of purchases $\mathbf{S}_i^{(t)}(k)$, where $\mathbf{S}_i^{(t)}(k)$ is the share of total expenditures allocated to expenditure category $k \in [1, 2, \ldots, K]$ and $K$ is the total number of COICOP expenditure categories, in region $i$ at time $t$. Thus, $\mathbf{S}_i^{(t)}(k) = (s_{i1}, s_{i2} \ldots, s_{ik})$ denotes the vector of the expenditures shares in each category made by individuals living in district area $i$, during the time interval $[t, t + w]$. We compared it with the average share of purchases in each COICOP category for all transactions in our dataset $\bar{\mathbf{S}}(k)$, which represents the average behavior of a consumer throughout the country. Finally, if a substantial divergence suddenly appeared in our dataset, we considered it a change in the consumption pattern of an individual living in the particular area. In this study, we limited the geographic area to the 42 districts of Lima. In (3), we define the divergence measure $\mathcal{D}^{(2)}$ as the average KLD between $\mathbf{S}_i^{(j)}$ the distribution of the share of purchases at time $j$ in district $j$ and $\mathbf{S}_i^{(k)}$ the distribution of share of purchases at time $k$ in district $j$ over $w$ days.

$$KLD(\mathbf{P} \parallel \mathbf{Q}) = \sum_{k \in K} \mathbf{P}(k) \, \log_2 \left[ \frac{\mathbf{P}(k)}{\mathbf{Q}(k)} \right] \tag{1}$$

$$\mathcal{D}_{(i,w)}^{(1)}(j) = KLD(\mathbf{S}_i^{(j)} \parallel \bar{\mathbf{S}}) \tag{2}$$

$$\mathcal{D}^{(2)}_{(w,i)}(j) = \frac{1}{w}\sum_{d=1}^{w} KLD\big(\mathbf{S}_i^{(j)} \,\|\, \mathbf{S}_i^{(k+d)}\big) \tag{3}$$

## Causal impact

Causal impact captures causality by measuring the difference between two different time series: one series under treatment, and another series not under treatment. Therefore, the causal inference algorithm takes three parameters: 1) the observed $o$ time series $\mathbf{T}^{(t)}_{o,i}(k)$, where $k \in [1, 2, \ldots, K]$ is the total number of COICOP expenditure categories, in region $i$ within period $t$; 2) a control time series $c$ $\mathbf{T}^{(t)}_{c,i}(k)$: and 3) an intervention date $d$, which is February 15, 2017, the start date of the El Niño phenomenon. The first step is to train a statistical or machine learning model, using parts of the time series before the intervention date (i.e., pre-period) to learn how to explain the studied time series $\mathbf{T}^{(t)}_{o,i}(k)$ as a function of the control time series $\mathbf{T}^{(t)}_{c,i}(k)$. Then, the learned model is used to predict the behavior of the studied time series after the intervention date (i.e., post-period), which provides the contrafactual estimate. Finally, the algorithm measures the difference between the predicted and real-time series to capture causal impact. It is worth noting that the model used in this study is the Bayesian structural time-series model [25].

## Individual stationary purchasing behavior

To capture individual purchasing behavior, we use a MMC. A MMC [36] models the mobility behavior of an individual as a discrete stochastic process in which the probability of moving to a state (i.e., point-of-interest (POI)) depends only on the previously visited state and the probability distribution of the transitions between states. In our case, POIs represent the merchants visited by clients (see Fig 11(a)). More precisely, a MMC is composed of a set of states $\{M_1, M_2, \cdots, M_N\}$ where $N$ is the total number of merchants, in which a transaction takes place. Transitions, such as $T^{(u)}_{M_i,M_j}(t)$, represent the probability of a user $u$ moving from state $M_i$ to state $M_j$ during the interval $t$ of 7 days (see Fig 11(b)). Finally, we computed the steady state probability vector $\pi^{(u)}(t)$ where each $\boldsymbol{\pi}^{(u)}_i(t)$ represents the probability of purchase of a product in merchant $i$ from the user $u$ during the $t$ period (see Fig 11(c)).

$$\boldsymbol{\Pi}^{(u)}_{\mathcal{K}}(t) = \sum_{i \in \mathcal{K}} \boldsymbol{\pi}^{(u)}_i(t) \tag{4}$$

$$\mathbf{Rel}^{(u)}(t) = \{\boldsymbol{\Pi}^{(u)}_{\mathcal{K}}(t) | \mathcal{K} \in [1, \ldots, 13], \ \text{with} \ \forall \mathcal{K}' > \mathcal{K}, \ \boldsymbol{\Pi}^{(u)}_{\mathcal{K}}(t) \leq \boldsymbol{\Pi}^{(u)}_{\mathcal{K}'}(t)\} \tag{5}$$

In our context, since we are more interested by the type of goods purchased than by the specific merchant, we aggregated in (4) the steady state vector $\boldsymbol{\Pi}_{\mathcal{K}}(t)$ that represents the probability of completing a purchase in a given COICOP category $\mathcal{K}$ (e.g., health, or clothing and footwear categories). In (5) we created a relevance metric set $\mathbf{Rel}^{(u)}(t)$ sorted in a descending order (see Fig 11(d)). Therefore we used the relevance metric to compute the discounted cumulative gain (DCG) in (6) to measure the consumption variation over time for individuals. The principle behind this metric is that COICOP categories $\mathcal{K}$ with higher probability $\boldsymbol{\Pi}_{\mathcal{K}}$ are more relevant. In (7) we capture the purchasing variation between consecutive periods for the same individual $u$. Finally, in (8) to capture the purchase variation for all individuals the mean

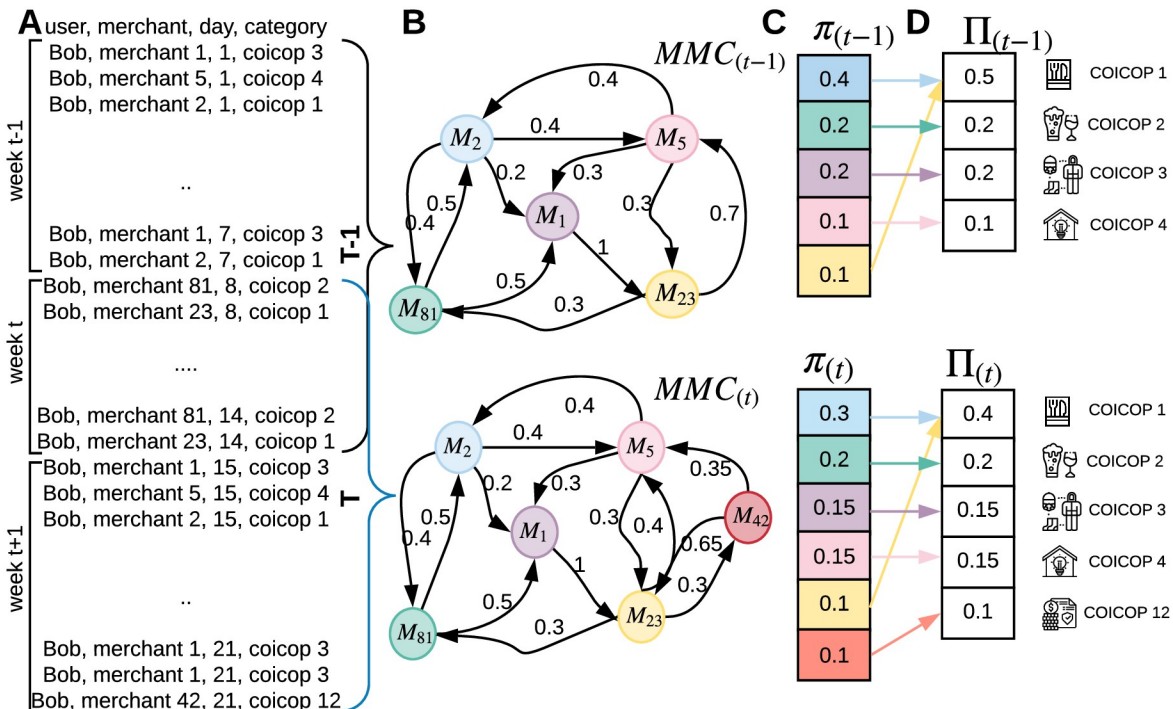

**Fig 11. Individual stationary purchasing behavior process.**

of all individuals' *nDGC* is computed for each period *t*.

$$DGC^{(u)}(t) = \sum_{i=1}^{|\mathcal{K}|} \frac{2^{\mathbf{Rel}_i^{(u)}(t)} - 1}{log_2(i+1)} \tag{6}$$

$$nDCG^{(u)}(t) = \frac{DGC^{(u)}(t)}{DGC^{(u)}(t-1)} \tag{7}$$

$$\overline{nDCG}(t) = \frac{1}{|u|} \sum_u nDGC^{(u)}(t) \tag{8}$$

## Transaction graph

Based on the transaction dataset, we can define the transaction graph $\mathcal{G}$ as a list of temporal snapshots $\mathcal{G} = [\mathbf{G}_{t_0}, \ldots, \mathbf{G}_{t_k}]$ where $\mathbf{G}_t(V, E(t), \mathbf{W}(t))$ is a weighted directed graph in which nodes $V$ represent the merchant. An edge $e_{ij}(t)$ exists if there is at least one credit or debit card holder that completed a purchase with merchant $i$ and then merchant $j$ during the interval [$t − 4, t + 4$] (in days). The weights represent $w_{ij}(t)$, the number of co-transactions made by different credit card holders during the interval interval [$t − 4, t + 4$] between merchant $i$ and merchant $j$. The orientation of the edge represents the temporal order of the purchase sequence. For instance, given the following purchase pattern of user $l$ during the interval [$t_0 − 4, t_0 + 4$]: $\{P_u^{(l)}(t_0 − 1), P_v^{(l)}(t_0 + 1), P_w^{(l)}(t_0 + 3)\}$, the directed edges, $(u, v)$ and $(v, w)$ would be present in graph in the snapshot $\mathbf{G}_{t_0}$.

## PageRank

PageRank quantifies the importance of nodes (centrality) in a network by computing the dominant eigenvector of the PageRank matrix (or Google matrix [37]). Using the PageRank algorithm we computed the ranking $c_i(t)$ (9) of all merchants $i$ in our dataset at each given instant $t$, where $\mathbf{W}(t)$ is the weighted adjacency matrix of the snapshot $\mathbf{G}_t$ of transaction graph $\mathcal{G}$.

$$\mathbf{C}(t) = \alpha \mathbf{S}(t)^{-1} \mathbf{W}(t)^T + (1 - \alpha) \frac{1}{N} \mathbf{1} \mathbf{1}^T \tag{9}$$

Here $s_{ii}(t) = \sum_{j=1}^{N} w_{ij}(t)$, $\mathbf{1} = [1, \ldots, 1]^T$ and $\alpha = 0.85$. We derived the evolution of the ranking of each merchant $r_i(t)$ by computing the PageRank of each snapshot $\mathbf{G}(t_0), \ldots, \mathbf{G}(t_k)$ of our dataset. Finally, we computed the normalized ranking $r_i(t) \in [0, 1]$ (10) of each merchant $i$ at time $t$ as follows:

$$r_i(t) = 1 - \frac{c_i(t) - 1}{\max_i c_i(t)} \tag{10}$$

In Fig 7, we can observe several examples of the PageRank evolution in time for distinct merchants.

## Time series clustering

To cluster the time series of the ranking evolution of each merchant, we used 1d-SAX [38] a method for representing a time series as a sequence of symbols containing information about the average and trend of the series on a segment. Symbolic aggregate approximation (SAX) is one of the main symbolization techniques for time series. Our goal was to cluster the merchant ranking evolution according to the merchants' behavior, especially during the main ENSO event of Feb. 2017. To do so, we used the 1d-SAX algorithm to help extract the main trends in each time series. The 1d-SAX algorithm is based on three main steps:

1. Divide the time series into segments of length $L$.

2. Compute the linear regression of the time series on each segment.

3. Quantize these regressions into a symbol from an alphabet of size $N$.

After the 1d-SAX transformation (see Fig 12 for different steps of the transformation of the time series), we clustered the time series using the standard **K-mean** algorithm using the Euclidean distance as the distance metric. In Fig 13 we depict the silhouette score of our clustering method across different parameters range of cluster numbers and segment lengths. Based on sensitivity analysis we determined that six clusters and a segment length $L = 15$ led to an effective trade-off for clustering our time series with an alphabet size of $N = 8$.

## Tracking the evolution of the core/periphery structure of the transaction graph

Many networks exhibit a core/periphery structure [40, 41], in which a set of nodes forms a densely connected group that governs the overall behavior of the network. This structure is recognized as a key mesoscale structure in complex networks that influences the functionality of a network, as demonstrated in the delivery of information in the Internet [42].

To partition nodes into two classes, core $V_c$ and periphery $V_p$, we used the method developed by Ma *et al.* [30]. The proposed method ranks the nodes by degree in descending order. For a given node, it divides its links into two groups: those with nodes of a higher rank and

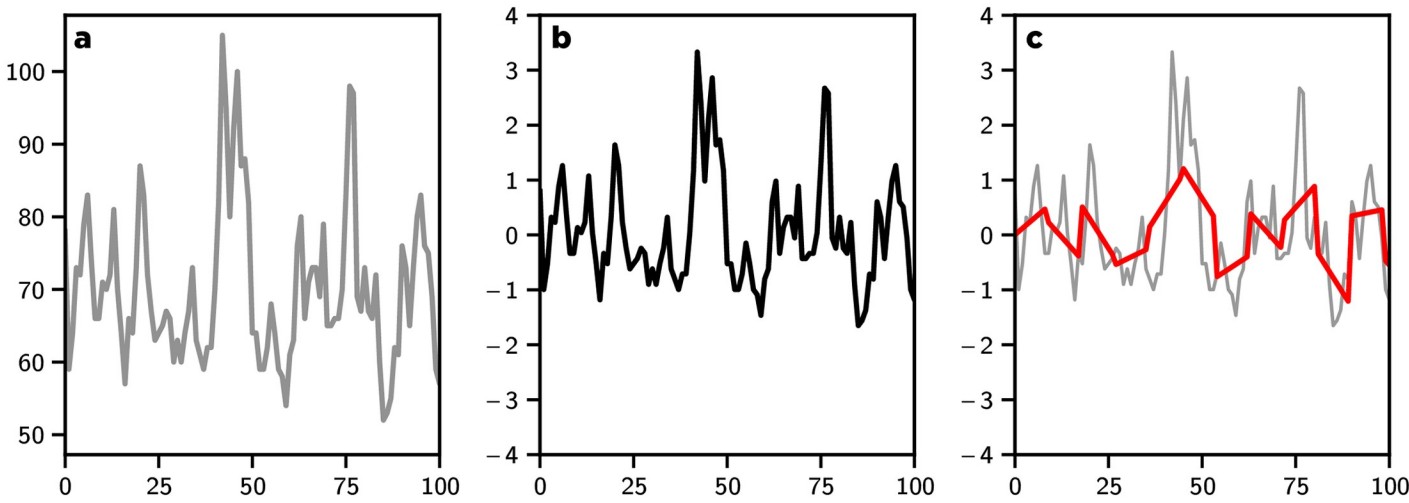

**Fig 12. Example of preprocessing step used to prepare the time series before the clustering step using the tslearn tool chain [39].** a) Row time series $r_i(t)$ of the rank of merchant $i$ over time. b) The time series $r_i(t)$ was standardized by subtracting the mean and dividing by the variance. c) The 1d-SAX transformation was applied to the standardized time series $r_i(t)$ with an alphabet size of $N = 8$.

those of a lower rank. More formally, a node of rank $r$ has degree $k_r$ the number of links it shares with nodes of a higher rank is $k_r^+$, and the number of links with nodes of a lower rank is $k_r - k_r^+$. To distinguish the core's node from the remaining nodes, we examined the nodes starting from the node of the highest rank toward the node of the lowest rank and stopped when we identified node $r^*$ where $k_r^+$ reached its maximum as depicted in Fig 14. Because the

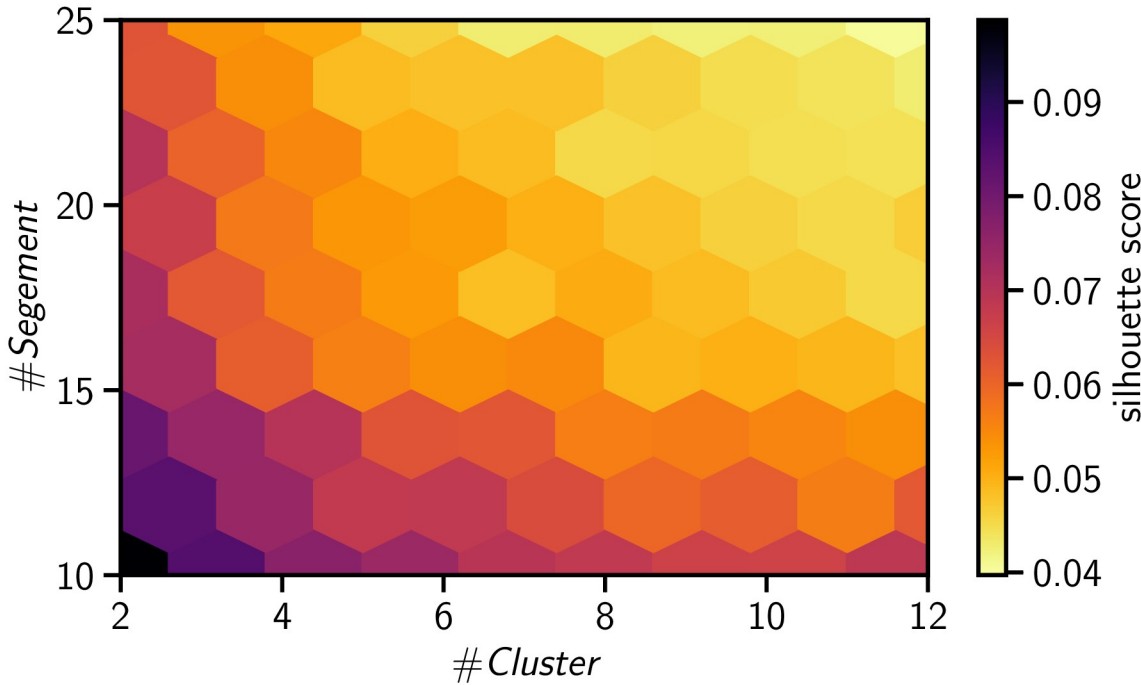

**Fig 13. Silhouette score.**

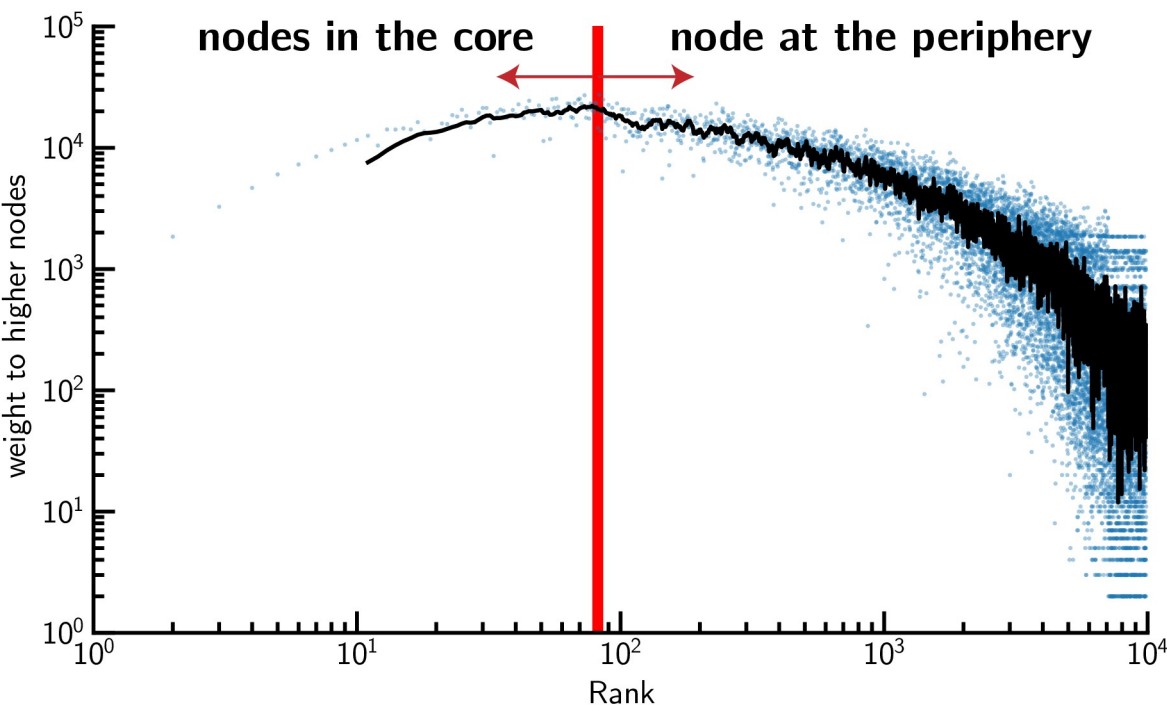

**Fig 14. Example of core/periphery detection for the weighted directed network.** Here the core/periphery of a slice of the transaction graph of May 2, 2017 is displayed. The core size is $|V_{core}(t)| = 82$ for a graph with $|V| = 11, 179$ vertices.

definition of a rich core can be extended to weighted directed networks, we used the method proposed in [30] to extract the core/periphery structure of all graph snapshots $\mathbf{G}_t$ (weighted and directed) of the full transaction graph $\mathcal{G}(V, E)$. Finally, we computed the size of the core $|V_c(t)|$ over time where $V_c(t) \subseteq V$.

## Supporting information

**S1 File.**
(PDF)

## Acknowledgments

This work was conducted at the Energy4Climate Interdisciplinary Center (E4C) of IP Paris and Ecole des Ponts ParisTech. The authors thank the computing resource platform of Telecom SudParis and M. Christian Bac for his help. The authors thank M. Latapy for many useful discussions.

## Author Contributions

**Data curation:** Hugo Alatrista-Salas, Vincent Gauthier, Miguel Nunez-del-Prado.

**Formal analysis:** Vincent Gauthier.

**Funding acquisition:** Hugo Alatrista-Salas, Vincent Gauthier, Miguel Nunez-del-Prado.

**Investigation:** Hugo Alatrista-Salas, Miguel Nunez-del-Prado.

**Methodology:** Hugo Alatrista-Salas, Vincent Gauthier, Miguel Nunez-del-Prado, Monique Becker.

**Resources:** Vincent Gauthier.

**Writing – original draft:** Hugo Alatrista-Salas, Vincent Gauthier, Miguel Nunez-del-Prado, Monique Becker.

**Writing – review & editing:** Hugo Alatrista-Salas, Vincent Gauthier, Miguel Nunez-del-Prado, Monique Becker.

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
