## [Decision Letter · Decision Letter 0]

22 Oct 2020

PONE-D-20-22088

Impact of natural disasters on consumer behavior:  case of the 2017 El Niño phenomenon  in Peru

PLOS ONE

Dear Dr. Alatrista-Salas,

Thank you for submitting your manuscript to PLOS ONE. After careful consideration, we feel that it has merit but does not fully meet PLOS ONE’s publication criteria as it currently stands. Therefore, we invite you to submit a revised version of the manuscript that addresses the points raised during the review process.

We look forward to receiving your revised manuscript.

Kind regards,

Jose M. Riascos, Ph.D.

Academic Editor

PLOS ONE

Journal Requirements:

3. We note that Figures 1, 2, 5, S3 in your submission contain map images which may be copyrighted. All PLOS content is published under the Creative Commons Attribution License (CC BY 4.0), which means that the manuscript, images, and Supporting Information files will be freely available online, and any third party is permitted to access, download, copy, distribute, and use these materials in any way, even commercially, with proper attribution. For these reasons, we cannot publish previously copyrighted maps or satellite images created using proprietary data, such as Google software (Google Maps, Street View, and Earth). For more information, see our copyright guidelines: http://journals.plos.org/plosone/s/licenses-and-copyright.

3.1.    You may seek permission from the original copyright holder of Figures 1, 2, 5, S3 to publish the content specifically under the CC BY 4.0 license. 

3.2.    If you are unable to obtain permission from the original copyright holder to publish these figures under the CC BY 4.0 license or if the copyright holder’s requirements are incompatible with the CC BY 4.0 license, please either i) remove the figure or ii) supply a replacement figure that complies with the CC BY 4.0 license. Please check copyright information on all replacement figures and update the figure caption with source information. If applicable, please specify in the figure caption text when a figure is similar but not identical to the original image and is therefore for illustrative purposes only.

Reviewers' comments:

Reviewer's Responses to Questions

**Comments to the Author**

1. Is the manuscript technically sound, and do the data support the conclusions?

Reviewer #1: Yes

Reviewer #2: Yes

2. Has the statistical analysis been performed appropriately and rigorously? 

Reviewer #1: Yes

Reviewer #2: Yes

3. Have the authors made all data underlying the findings in their manuscript fully available?

Reviewer #1: Yes

Reviewer #2: Yes

4. Is the manuscript presented in an intelligible fashion and written in standard English?

Reviewer #1: Yes

Reviewer #2: Yes

5. Review Comments to the Author

Reviewer #1: It is a well-organized paper. It covers understudied areas of disaster resiliency. The findings are thoroughly discussed and conclusions were generated based on findings. Your paper is good to publish.

Reviewer #2: El Nino Southern Oscillation (ENSO) is a climatic phenomenon consisting of a temperature increase in the equatorial Pacific area. At the end of 2016 and in early 2017, ENSO had an abrupt change leading to heavy rains and floods. It is possible to see higher frequency of such atypical phenomenon due to climate change in the future. However, it remains unclear how these weather anomaly affects the consumption behavior of people as well as the resilience of retail structures.

This paper conduct a multi-scale analysis of the consumption patterns based on a credit and debit card transaction dataset of roughly million Peruvian citizens gathered over a 2-year period from 2016 to 2017. The authors find that, at the macro-level, there was a slowdown of economic activities triggered by ENSO events but the overall economic activities recovered swiftly from the events. At the micro-level, the consumption of necessities increased but the non-necessities decreased. In addition, the core network structure of the transaction graph observed a reduction in the size of the core network structure during ENSO events.

This paper contributes to the literature by documenting how consumption behaviors, from both macro and micro level, respond to the weather extreme events. The findings are useful to policy design that aims to mitigate the negative impacts of climatic events and improve the effectiveness of recovery efforts. Below are some comments and suggestions that I hope are useful to the authors.

First, this paper use the Kullback-Leibler divergence (KLD) method to capture an anomalous change in the purchase distribution of the population (Figure 3&4). It is not straightforward enough that these changes of consumption patterns are results of ENSO. Since the El Nino is characterized by heavy rains and floods, it is useful to replicate the current figures but replace the vertical axis with rainfall or floods. We expect to see similar patterns of rainfall as in consumption. In addition, it is also useful to replicate the analysis using data from years without El Nino costero. We expect to see no previous patterns as in years with El Nino costero. Also, it is not clear whether the seasonal pattern as well as time trends of consumption are adjusted.

Second, in the section of Causality Analysis of the ENSO, this paper discovered three modes: districts were negatively impacted, districts that continued to function as usual, and districts that experienced an increase in purchases. More discussions are needed to explain why the impacts in different regions can be postive impact, negative and netural? How are these findings connected with rainfall or floods due to the ENSO. Answering these questions help disentangle the impacts from the ENSO instead of other economic confounders.

Third, since the data contain information social-economic factors, it would be interesting to separately estimate the impacts for cohorts with different social-economic factors. For example, does the consumption behaviors respond differently to the weather events for the male and female? How does the weather extremes affects the younger and older cohorts differently?

6. PLOS authors have the option to publish the peer review history of their article (what does this mean?). If published, this will include your full peer review and any attached files.

Reviewer #1: No

Reviewer #2: No

---

## [Author Response · Author response to Decision Letter 0]

7 Dec 2020

Please, find a reply to the specific reviewers' comments in the uploaded file "Response to reviewers.pdf."

---

## [Editor Report · Decision Letter 1]

9 Dec 2020

Impact of natural disasters on consumer behavior:  case of the 2017 El Niño phenomenon  in Peru

PONE-D-20-22088R1

Dear Dr. Alatrista-Salas,

We’re pleased to inform you that your manuscript has been judged scientifically suitable for publication and will be formally accepted for publication once it meets all outstanding technical requirements.

Kind regards,

Jose M. Riascos, Ph.D.

Academic Editor

PLOS ONE
---

## [Editor Report · Acceptance letter]

17 Dec 2020

PONE-D-20-22088R1 

Impact of natural disasters on consumer behavior:  case of the 2017 El Niño phenomenon  in Peru 

Dear Dr. Alatrista-Salas:

I'm pleased to inform you that your manuscript has been deemed suitable for publication in PLOS ONE. Congratulations! Your manuscript is now with our production department. 

Kind regards, 

on behalf of

Professor Jose M. Riascos 

Academic Editor

PLOS ONE